# New Canadian and Provincial Records of Coleoptera Resulting from Annual Canadian Food Inspection Agency Surveillance for Detection of Non-Native, Potentially Invasive Forest Insects

**DOI:** 10.3390/insects13080708

**Published:** 2022-08-06

**Authors:** Graham S. Thurston, Alison Slater, Inna Nei, Josie Roberts, Karen McLachlan Hamilton, Jon D. Sweeney, Troy Kimoto

**Affiliations:** 1Canadian Food Inspection Agency, 960 Carling Avenue, Building 18, Ottawa, ON K1A 0Y9, Canada; 2Canadian Food Inspection Agency, 506 West Burnside Road, Victoria, BC V8Z 4N9, Canada; 3Natural Resources Canada, Canadian Forest Service—Atlantic Forestry Centre, P.O. Box 4000, Fredericton, NB E3B 5P7, Canada; 4Canadian Food Inspection Agency, 4321 Still Creek Drive, Burnaby, BC V5C 6S7, Canada

**Keywords:** Coleoptera, non-indigenous species, woodborers, surveillance, trapping, invasive insects

## Abstract

**Simple Summary:**

Early detection of adventive and potentially invasive insects that may threaten the health of forests is essential for their successful management in Canada and the world at large. To that end, the Canadian Food Inspection Agency (CFIA) conducts annual surveys at sites with high risk of adventive species introductions (e.g., ports, industrial zones, disposal sites for solid wood packaging material) using semiochemical-baited traps and also by rearing insects from trunk sections collected from stressed trees. We report 31 new Canadian provincial records of beetle species detected in surveys conducted from 2011 to 2021, including 13 new records for Canada and 9 species adventive to North America. Nine of the new Canadian records were native North American species previously detected only south of the border. All but three species belong to the Curculionidae family (“snout beetles”) and most of these were in the subfamily Scolytinae (bark and ambrosia beetles). Rearing of insects from trunk sections of stressed trees accounted for two new species records whereas trapping accounted for the remainder. These surveys not only assist our efforts to manage forest insects by documenting new species introductions and apparent range expansions but also increase our knowledge of biodiversity.

**Abstract:**

The arrival and establishment of adventive, invasive forest insects are a threat to the health, diversity, and productivity of forests in Canada and the world at large, and their early detection is essential for successful eradication and management. For that reason, the Canadian Food Inspection Agency (CFIA) conducts annual surveys at high risk sites such as international ports and freight terminals, industrial zones, and disposal sites for solid wood packaging material using two methods: (1) semiochemical-baited traps deployed in a total of about 63–80 sites per year in British Columbia (BC), Ontario (ON), Quebec (QC), New Brunswick (NB), Nova Scotia (NS), and Newfoundland and Labrador (NL); and (2) rearing of insects from bolts collected from stressed trees and incubated in modified shipping containers in four cities (Vancouver, Toronto, Montreal, and Halifax). We report 31 new Canadian provincial records of Coleoptera from surveys conducted in 2011–2021, including 13 new records for Canada and 9 species adventive to North America (indicated by †). Nine of the new Canadian records were native North American species previously detected only south of the border. All but three species belong to the Curculionidae family and most of these were in the subfamily Scolytinae. The records include: *Xenomelanophila miranda* (LeConte) (Canada, BC) (Buprestidae: Buprestinae); *Neoclytus mucronatus mucronatus* (Fabricius) (BC) (Cerambycidae: Cerambycinae); *Amphicerus cornutus* (Pallas) (Canada, BC) (Bostrichidae: Bostrichinae); *Mecinus janthinus* (Germar)† (ON) (Curculionidae: Curculioninae); *Aulacobaris lepidii* (Germar)† (Canada, ON); *Buchananius striatus* (LeConte) (ON) (Curculionidae: Baridinae); *Cylindrocopturus binotatus* LeConte (Canada, ON) (Curculionidae: Conoderinae); *Himatium errans* LeConte (ON); *Phloeophagus canadensis* Van Dyke (ON); *Rhyncolus spretus* Casey (Canada, BC); *Stenomimus pallidus* (Boheman) (Canada, ON); *Tomolips quercicola* (Boheman) (Canada, ON) (Curculionidae: Cossoninae); *Strophosoma melanogrammum* (Forster)† (NB) (Curculionidae: Entiminae); *Conotrachelus aratus* (Germar) (ON) (Curculionidae: Molytinae); *Anisandrus maiche* Stark† (Canada, ON, QC); *Cnesinus strigicollis* LeConte (Canada, ON); *Cyclorhipidion pelliculosum* (Eichhoff)† (Canada, ON, QC); *Hylesinus fasciatus* LeConte (QC); *Hylesinus pruinosus* Eichhoff (QC); *Hypothenemus interstitialis* (Hopkins) (Canada, ON); *Lymantor alaskanus* Wood (BC); *Pityogenes bidentatus* (Herbst)† (Canada, ON); *Scolytus mali* (Bechstein)† (BC); *Scolytus schevyrewi* Semenov† (QC); *Trypodendron scabricollis* (LeConte) (Canada, ON); *Trypophloeus populi* Hopkins (QC); *Xylechinus americanus* Blackman (NFLB); and *Xylosandrus crassiusculus* (Motschulsky)† (BC, QC) (Curculionidae: Scolytinae). We also provide additional data confirming the presence of the adventive *Hylastes opacus* Erichson† in NS. Rearing of insects from bolts accounted for two new records (*H. pruinosus*, *R. spretus*) and trapping accounted for the remainder. These surveys not only assist our efforts to manage forest insects by documenting new species introductions and apparent range expansions but also increase our knowledge of biodiversity.

## 1. Introduction

The introduction and establishment of bark- and wood-boring beetles outside of their native ranges is an ongoing threat to many of the world’s forests as some of these species become damaging invasive pests [1,2] with significant impacts on forest health and the economy [3,4]. The emerald ash borer, *Agrilus planipennis* (Buprestidae: Agrilinae) in North America [5] and European Russia [6], the red turpentine beetle, *Dendroctonus valens* (Curculionidae: Scolytinae) in China [7], and the Asian longhorned beetle, *Anoplophora glabripennis* (Cerambycidae: Lamiinae), in North America and Europe [8] are just a few examples of highly damaging invasive species established outside of their native range. The most likely invasion pathway for most bark- and wood-boring beetles is in solid wood packaging material, but other common pathways for non-native forest insects include importation of live plants, wooden handicrafts, “hitchhiking” (e.g., accidental stowaways in luggage or onboard ships), and intentional introductions of biological control agents [1,9]. The international adoption of measures for phytosanitary treatment of solid wood packaging [10] has reduced but not eliminated the interception of live insects in shipments [11] and global rate of establishment of species outside of their native range has not leveled off [12]. For this reason, plant protection organizations in many countries conduct surveillance programs to detect new introductions and monitor the ranges of native and adventive species [13,14,15,16,17].

As Canada’s national plant protection organization, the Canadian Food Inspection Agency (CFIA) has the mandate to prevent the introduction and establishment in Canada of plant pests of quarantine significance, and to monitor and limit the spread of those organisms within Canada. To fulfill this mandate, the CFIA conducts a variety of pest surveys, which include annual forest pest trapping and rearing surveys for detection of introduced forest pests. Traps baited with plant volatiles and/or insect pheromones are commonly used for surveillance of non-native forest insects in North America and abroad [18] and have resulted in the discovery of several species established outside of their native range [16,19,20]. These surveys are intended to detect potentially invasive pests as early as possible because the probability of eradication or containment decreases and management costs increase as the size of the infested area increases [21]. These surveys also provide valuable information on the range expansions of insects within North America and Canada, and much needed information on the diversity of Canada’s insect fauna. New Canadian and provincial coleopteran records from these CFIA forest insect surveys dating back to 2011 are reported herein.

## 2. Methods and Materials

*Trapping surveys.* The forest pest trapping survey is a commodity pathway survey, targeting non-indigenous wood-boring insects that could be introduced into Canada through international solid wood packaging material (SWPM) and wooden handicrafts. The CFIA conducts this survey in 63–80 sites per year in the provinces of British Columbia, Ontario, Quebec, New Brunswick, Nova Scotia, and Newfoundland and Labrador. Survey sites are those deemed high risk for adventive species introductions, such as industrial and commercial zones, international ports and freight terminals, industrial and municipal disposal facilities (landfills), and SWPM disposal facilities. Cities and sites with greatest volume of international trade and greatest risk of adventive species introductions are surveyed every year, whereas areas with moderate risk are surveyed periodically on a rotational basis. Twelve-unit, black multiple funnel (Lindgren) traps [22] (Appendix A) are placed in treed or forested areas within 5 km of high-risk sites. From the early 2000s to 2011, the three lure combinations used were: (1) ultra-high release (UHR) ethanol and UHR (-)-alpha-pinene; (2) *cis*-verbenol, racemic ipsdienol and 2-methyl-3-buten-2-ol (i.e., exotic bark beetle lure); (3) UHR ethanol. Traps with the first two lure combinations (three traps per lure combination, six traps per site) were placed between conifers whereas traps baited with ethanol alone (three traps per site) were placed between deciduous or coniferous trees. 

Beginning in 2012, in addition to using traps baited with UHR ethanol and UHR alpha-pinene, CFIA began baiting traps at some survey sites with a three-lure combination of UHR ethanol plus racemic 3-hydroxyhexan-2-one (C6-ketol) and 3-hydroxyoctan-2-one (C8-ketol), which contain sex-aggregation pheromones attractive to several species of longhorn beetles primarily in the subfamily Cerambycinae [23,24] (Appendix A). Traps included an ethanol lure because it enhances attraction of several longhorn beetle species to their pheromones [25] and is attractive to many species of Scolytinae by itself [26,27]. From 2015 to 2021, traps were baited with either a three-lure combination of: (1) racemic (*E*,*Z*)-fuscumol [(*E*,*Z*)-6,10-dimethyl-5,9-undecadien-2-ol], racemic (*E*,*Z*)-fuscumol acetate [(*E*,*Z*)-6,10-dimethyl-5,9-undecadien-2-yl acetate] and UHR ethanol (i.e., general longhorn lure); or a four-component combination of racemic ipsenol, monochamol (2-undecyloxy-1-ethanol), UHR (-)-alpha-pinene and UHR ethanol (i.e., *Monochamus* lure). Fuscumol and its acetate are pheromones of some species of Lamiinae and Spondylidinae [28,29,30]. Monochamol is the aggregation pheromone for many *Monochamus* species [31,32,33], while ipsenol, alpha-pinene and ethanol not only enhance attraction of *Monochamus* to monochamol [34], but also attract many species of Scolytinae [26]. Except for monochamol and ipsenol, which are released from the same release device, each lure component was in a separate release device. The UHR ethanol and UHR (-)-alpha-pinene lures were suspended on the outside of the trap and the pheromone lures were placed inside the funnel trap, 2–3 funnels up from the collecting cup (Appendix A). Traps baited with the general longhorn lure were placed between either coniferous or deciduous trees whereas traps baited with the *Monochamus* lure were placed between coniferous trees. 

Traps were hung from a polypropylene rope tied between two trees, with the collection jar at least 30 cm above ground level or above understory vegetation. In addition, 12-unit green funnel traps were placed in tree canopies in Richmond, British Columbia in 2016 as part of a pilot project (Appendix A). In 2018, the three Quebec offices piloted placement of green funnel traps in tree canopies on a larger operational scale. From 2011 to 2017, undiluted propylene glycol, denatonium benzoate and unscented soap was placed in the collection cup of each trap to preserve specimens. However, beginning in 2017 in Quebec and in 2018 in Ontario, a saturated salt solution was used instead of propylene glycol because it is cheaper and just as effective for preserving specimens. Traps were placed in the field between March and October and samples were collected every 2–4 weeks. Samples were taken back to the office, cleaned with cool running water, placed in vials with non-denatured ethanol and sent to CFIA’s ISO 17025 accredited entomology diagnostic laboratory for identification.

*Rearing surveys.* As many wood-boring insects are not attracted to the semiochemicals used in our trapping survey, a forest pest rearing survey was also conducted to increase the likelihood of detecting new introduced species. This rearing survey was conducted in the City of Montreal, QC, Halifax Regional Municipality, NS, and various municipalities within greater Vancouver, BC, and greater Toronto, ON, from 2006 to 2021. Trees targeted by the city’s hazard tree removal programs that occurred in high-risk sites (e.g., near landfills, industrial and commercial zones) and that exhibited signs of infestation (e.g., exit holes, oviposition niches, exposed larval galleries, etc.) were selected. Two bolts from each selected tree were placed in mobile rearing facilities, and emerging insects were regularly collected for identification as per Bullas-Appleton et al. [14]. Bolts were retained for up to four years. Bolts from a wide variety of angiosperm and gymnosperm trees in these four regions were placed in the rearing facilities.

*Species identification.* Species were identified by highly trained personnel (authors AS, IN, JR, KM), based on morphological characters using published keys and other reference material [19,35,36,37,38,39,40,41,42,43,44,45,46,47,48,49,50,51]. Any specimens for which species determination was not certain, were examined and confirmed by taxonomists who specialize in Cerambycidae, Curculionidae and Buprestidae at Agriculture and Agri-Food Canada’s Canadian National Collection of Insects, Arachnids and Nematodes, Ottawa, Ontario (CNC), the Canadian Museum of Nature, Gatineau, Quebec, and the Invasive Species Centre, Sault Ste. Marie, Ontario. Voucher specimens have been deposited in the CNC, the main CFIA entomology collection, Ottawa, Ontario (CFIA) and the regional CFIA entomology collection in Victoria, BC (CFIA-BC).

**Distribution.** All species are cited with current distribution in Canada and Alaska, using abbreviations for the states, provinces, and territories (Table 1). Regional county municipalities (MRC) or equivalent territories (ET) are provided for Quebec records, and county records are provided for other provinces. New provincial records are indicated in **bold** under **Distribution in Canada and Alaska**. 

## 3. Results

We report 31 new provincial records of Coleoptera from BC (7), ON (15), QC (7), NB (1), and NL (1), 13 of which are also new Canadian records. We also confirm the presence of *Hylastes opacus* in Nova Scotia, previously reported by Webster et al. [52]. Ten of the detected species (and 9 of the new records) are adventive to Canada and are indicated by † (Table 2). For each new record we provide notes on the species’ historic and current distribution, known hosts, and detailed specimen collection data, including: province, county or regional district, town or city, geographical coordinates in decimal degrees, collection date, name of collector, method of collection (i.e., type of lure, color and height of the trap), number of voucher specimens, and the insect collection in which these voucher specimens have been deposited (e.g., CNC, 1).


**BUPRESTIDAE: Buprestinae**

***Xenomelanophila miranda* (LeConte 1854), new to Canada and British Columbia.**


**Note:** Native to North America. Range: western USA and Mexico. Host: *Juniperus* spp. Oviposits on smoldering wood [53].

**Specimen data: British Columbia, Central Okanagan Regional District**, Kelowna, 49.94681, −119.41285, 17 August 2016, coll: J. Dekker, black Lindgren funnel trap at ground level baited with *Monochamus* lure (CFIA-BC 1).


**Distribution in Canada and Alaska: BC.**



**CERAMBYCIDAE: Cerambycinae**

***Neoclytus mucronatus mucronatus* (Fabricius 1775), new to British Columbia.**


**Note:** Native to North America. Range: eastern USA; in Canada previously known from ON [54] and NB [55]. Hosts: *Celtis*, *Carya*, rarely *Pinus* spp. [56].

**Specimen data: British Columbia, Metro Vancouver Regional District**, Surrey, 49.18443, −122.81808, 21 August 2012, coll: T. Graham, black Lindgren funnel trap at ground level baited with UHR ethanol + C6 + C8 ketol lures (CFIA 1).

**Distribution in Canada and Alaska: BC**, ON, NB.


**BOSTRICHIDAE: Bostrichinae**

***Amphicerus cornutus* (Pallas 1772), new to Canada and British Columbia.**


**Note:** Native to North and South America. Range: South America, Caribbean, Mexico, widespread in USA. Can infest dry wood, timber, wood packaging material. Attacks many genera of dead or dying plants and trees [57].

**Specimen data: British Columbia, North Okanagan Regional District**, Vernon, 50.22030, −119.30270, 7 September 2011, coll: L. Ivey, black Lindgren funnel trap at ground level baited with UHR ethanol lure (CFIA 1).


**Distribution in Canada and Alaska: BC.**



**CURCULIONIDAE: Curculioninae**

***Mecinus janthinus* (Germar 1821) †, new to Ontario.**


**Note:** Native to Europe. Range: central and southern Europe, western North America. Previously known from BC, AB, QC, and NS [56]. Host: *Linaria* spp. Species introduced for biological control of toadflax, *Linaria* spp. (Plantaginaceae) [58].

**Specimen data: Ontario, Hastings County,** Belleville, 44.19116, −77.384370, 16 May 2018, coll: E. Reichert, black Lindgren funnel trap at ground level baited with general longhorn lure (CNC 1).

**Distribution in Canada and Alaska:** BC, AB, **ON,** QC, NS.


**CURCULIONIDAE: Baridinae**

***Aulacobaris lepidii* (Germar 1824) †, new to Canada and Ontario.**


**Note:** Native to the Palearctic; adventive in northeastern USA. Host: Brassicaceae; not known to be a serious pest [59].

**Specimen data: Ontario, York Regional Municipality,** Richmond Hill, 43.85606, −79.394100, 22 July 2019, coll: P. Sadowski, black Lindgren funnel trap baited with general longhorn lure (CNC 1).


**Distribution in Canada and Alaska: ON.**



***Buchananius striatus* (LeConte 1876), new to Ontario.**


**Note:** Native to North America. Range: eastern USA south to Mexico. Recently reported from QC [60]. Host: unknown; congener *B. sulcatus* has been recorded developing in fungal fruiting bodies [61].

**Specimen data: Ontario, Frontenac County,** Kingston, 44.27009, −76.530470, 7 July 2017, coll: E. Reichert, black Lindgren funnel trap at ground level baited with general longhorn lure (CNC 1).

**Distribution in Canada and Alaska: ON,** QC.


**CURCULIONIDAE: Conoderinae**

***Cylindrocopturus binotatus* (LeConte 1876), new to Canada and Ontario.**


**Note:** Native to North America. Range: eastern USA from New Jersey and New York south to Georgia, west to Ohio and Texas [40,62]. Fifteen specimens were collected in southern Ontario in 2020 with all but one specimen collected in black Lindgren funnel traps. Hosts: unknown, but most species in this genera feed in species of Compositae [63].

**Specimen data: Ontario, York Regional Municipality,** Toronto, 43.71915, −79.351020, 24 August 2020, coll: T. Peony, one specimen in black Lindgren funnel trap at ground level baited with general longhorn lure (CNC 1); Vaughan, 43.81893, −79.498190, 5 August 2020, coll: L. Shaw, one specimen in black Lindgren funnel trap at ground level baited with general longhorn lure (CNC 1); **Regional Municipality of Hamilton-Wentworth,** Hamilton, 43.25380, −79.901540, 10 July 2020, coll: M. Pearce, three specimens in black Lindgren funnel trap at ground level baited with general longhorn lure (CFIA 1), 7 August 2020, coll: C. Balardo, seven specimens in black Lindgren funnel trap at ground level baited with general longhorn lure (CNC 1; CFIA 2), 26 August 2020, coll. C. Balardo, two specimens in black Lindgren funnel trap at ground level baited with general longhorn lure (CFIA 2); Stoney Creek, 43.23243, −79.716160, 26 October 2020, coll: N. Mielewczyk, one specimen found in plastic wrapped aluminum in a windows and doors plant (CNC 1);


**Distribution in Canada and Alaska: ON**



**CURCULIONIDAE: Cossoninae**

***Himatium errans* LeConte 1876, new to Ontario.**


**Note:** Native to North America. Range: eastern Canada and USA. Known from QC, NB, and NS [56]. Hosts: Adults have been found under bark and in galleries of *Ips* on *Pinus* and have been reared from dead *Acer* branches [40,64].

**Specimen data: Ontario, Wellington County,** Guelph, 43.50011, −80.203930, 2 July 2019, coll: M. Pearce, black Lindgren funnel trap at ground level baited with *Monochamus* lure (CFIA 1).

**Distribution in Canada and Alaska: ON,** QC, NB, NS.


***Phloeophagus canadensis* Van Dyke 1927, new to Ontario.**


**Note:** Native to North America. Range: transcontinental in northern USA and Canada. Known from BC, AB, SK, MB, QC, NB [56]. Host: *Populus* [50,65].

**Specimen data: Ontario, York Regional Municipality,** Newmarket, 44.06099, −79.458520, 15 July 2019, coll: P. Sadowski, black Lindgren funnel trap at ground level baited with general longhorn lure (CNC 1).

**Distribution in Canada and Alaska:** AK, BC, AB, SK, MB, **ON,** QC, NB.


***Rhyncolus spretus* Casey 1892, new to Canada and British Columbia.**


**Note:** Native to North America. Range: western USA. Hosts: under bark and in decaying wood of *Corylus, Sambucus* and other hardwood species, reared from *Acer negundo* [65].

**Specimen data: British Columbia, Metro Vancouver Regional District,** New Westminster, 49.21756, −122.89634, 30 November 2016, coll: K. Heim, reared from *Acer* sp. (CFIA 2).


**Distribution in Canada and Alaska: BC.**



***Stenomimus pallidus* (Boheman 1845), new to Canada and Ontario.**


**Note:** Native to North America. Range: eastern and central USA. Hosts: *Carya* and *Juglans* spp. Possible vector of thousand cankers disease, *Geosmithia morbida* [66].

**Specimen data: Ontario, Oxford County,** Ingersoll, 43.00447, −80.902210, 21 June 2018, coll: P. Dolan, black Lindgren funnel trap at ground level baited with general longhorn lure (CNC 1).


**Distribution in Canada and Alaska: ON.**



***Tomolips quercicola* (Boheman 1845), new to Canada and Ontario.**


**Note:** Native to North America. Range: eastern USA to Central America. Hosts: wet wood, leaf litter, can infest damp wood of many species [40].

**Specimen data: Ontario, Regional Municipality of Niagara Region,** St. Catharines, 43.21646, −79.216660, 28 June 2018, coll: K. Cantera, black Lindgren funnel trap at ground level baited with general longhorn lure (CNC 1).


**Distribution in Canada and Alaska: ON.**



**CURCULIONIDAE: Entiminae**

***Strophosoma melanogrammum* (Forster 1771) †, new to New Brunswick.**


**Note:** Native to Europe, adventive in North America. Range: northeastern and northwestern North America; Europe. Previously known from BC, ON, QC, NS, PE, and NL [54]. Hosts: Adults feed on foliage of variety of both deciduous and coniferous trees and shrubs [39]. They are considered economic pests in Denmark due to cosmetic damage caused by adults feeding on needles of *Abies procera* used for production of ornamental greenery [67].

**Specimen data: New Brunswick, Saint John County,** Saint John, 45.28857, −66.056900, 20 June 2016, coll: N. Labbe, black Lindgren funnel trap at ground level baited with general longhorn lure (CNC 1).

**Distribution in Canada and Alaska:** BC, ON, QC, **NB,** NS, PE, NL.


**CURCULIONIDAE: Molytinae**

***Conotrachelus aratus* (Germar 1824), new to Ontario.**


**Note:** Native to North America. Range: eastern North America. Previously known from QC [56]. Hosts: Pest of *Carya* spp.; larvae tunnel in shoots and may cause early leaf drop or dieback [40,68].

**Specimen data: Ontario, Halton Regional Municipality,** Georgetown, 43.65808, −79.907170, 16 August 2018, coll: I. Bhanot, black Lindgren funnel trap at ground level baited with general longhorn lure (CFIA 1).

**Distribution in Canada and Alaska: ON,** QC.


**CURCULIONIDAE: Scolytinae**

***Anisandrus maiche* Stark 1936 †, new to Canada, Ontario and Quebec.**


**Note:** The first record of *A. maiche* in Canada was from an Ontario Ministry of Natural Resources trapping program in 2013. This detection has remained unreported until now. Native to the Palearctic and recently introduced to North America [19]. Range: Asia; eastern USA. Hosts: *Acer*, *Alnus*, *Corylus*, *Betula*, *Euonymus, Fraxinus*, *Juglans, Magnolia, Syringa, Phellodendron, Ulmus* [19,42]. Rabaglia et al. [19] state the potential impact that *A. maiche* may have in North America is difficult to predict. Although many species in the same tribe (Xyleborini) attack weakened trees, some species will attack wounded healthy trees. It is also possible that the symbiotic ambrosia fungi the beetles carry, though benign to hosts in their native range, may become pathogenic on novel hosts in North America, e.g., the adventive *Xyleborus glabratus* on Lauraceae in the southeastern USA [19,69]

**Specimen data: Ontario, Middlesex Co.,** London, 42.92696, −81.330924, 9 July 2013, coll: R. Lidster, black Lindgren funnel trap at ground level baited with 3-methyl-2-buten-1-ol (CFIA 1). **Quebec, TE de Longueuil,** Boucherville, 45.56404, −73.416660, 16 July 2018, coll: S. Lamontagne, black Lindgren funnel trap at ground level baited with general longhorn lure, (CFIA 1); Brossard, 45.45793, −73.429660, 16 &17 July 2018, coll: S. Lamontagne, green Lindgren funnel trap in tree canopy baited with general longhorn lure (CNC 2); black Lindgren funnel trap in canopy baited with general longhorn lure (CNC 2).


**Distribution in Canada and Alaska: ON, QC.**



***Cnesinus strigicollis* LeConte 1868, new to Canada and Ontario.**


**Note.** Native to North America. Range: eastern US; Mexico. Hosts: Attacks small stems and woody twigs of numerous hardwood tree species [51].

**Specimen data. Ontario, Essex Co.,** Windsor, 42.26563, −83.079990, 23 June 2016, coll: A. Morden, black Lindgren funnel trap at ground level baited with general longhorn lure (CFIA 1, CNC 3).


**Distribution in Canada and Alaska: ON**
**.**



***Cyclorhipidion pelliculosum* (Eichhoff 1878) †, new to Canada, Ontario and Quebec.**


**Note.** Native to Asia. Range: eastern North America; Eurasia. First detected in eastern US in 1987 [36,42], and reported then as *Xyleborus pelliculosus*. Hosts: *Acer*, *Quercus*, and *Alnus*.

**Specimen data. Ontario, Waterloo Co.,** Cambridge, 43.36742, −80.269530, 24 May 2017, coll: J. Maloney, black Lindgren funnel trap at ground level baited with general longhorn lure (CFIA 1). **Quebec****, MRC l’Érable,** Plessisville, 46.23704, −71.782940, 12 June 2017, coll: A. Deschênes, black Lindgren funnel trap at ground level baited with general longhorn lure (CFIA 1).


**Distribution in Canada and Alaska: ON, QC.**



***Hylastes opacus* Erichson 1836 †, new supporting data for Nova Scotia.**


**Note.** Native to Palearctic region. Range: throughout Eurasian conifer forests; southern Asia; South Africa; eastern North America. First recorded in North America in 1989 [70]. In Canada, previously known from ON, NB, QC, and BC [54]. First reported in Nova Scotia by Webster et al. [52]. Hosts: Primarily breeds in stumps and roots of dead or dying *Pinus* [71].

**Specimen data. Nova Scotia, Halifax Co.,** Middle Musquodoboit, 45.05312, −63.165370, 7 June 2016, coll: J. Young, black Lindgren funnel trap at ground level baited with C6 ketol + C8 ketol + UHR ethanol lure (CNC 1).

**Distribution in Canada and Alaska.** BC, ON, QC, NB, NS.


***Hylesinus fasciatus* LeConte 1868; new to Quebec.**


**Note.** Native to North America. Range: eastern North America. Known previously from ON [51]. Host: *Fraxinus*, considered to have risk of endangerment due to the destruction of ash trees by the emerald ash borer [72].

**Specimen data. Quebec, TE de Sherbrooke,** Sherbrooke, 45.40162, −71.975520, 2 June 2017, coll: S. Drouin, black Lindgren funnel trap at ground level baited with *Monochamus* lure, (CFIA 1, CNC 1).

**Distribution in Canada and Alaska:** ON, **QC.**


***Hylesinus pruinosus* Eichhoff 1868; new to Quebec.**


**Note.** Native to North America. Range: eastern North America. Known previously from ON [51]. Host: *Fraxinus* as only known host genus. This species is considered to have a high risk of endangerment over its entire range due to the destruction of ash trees by the emerald ash borer [72].

**Specimen data. Quebec, MRC de Beauharnois-Salaberry,** Salaberry-de-Valleyfield, 45.25158, −74.148040, 8 June 2017, coll: B. Bourbeau, reared from two *Fraxinus* bolts (CNC 3).

**Distribution in Canada and Alaska:** ON, **QC.**


***Hypothenemus interstitialis* (Hopkins 1915), new to Canada and Ontario.**


**Note.** Native to the Americas. Range: southeastern USA with a few distribution records reaching north to Illinois [73], south to Brazil and Peru. Hosts: many hardwood species native to Canada such as *Acer*, *Carya*, *Fagus*, and *Quercus*. Attacks injured or damaged twigs or small branches and breeds in the pith and xylem of small stems [51].

**Specimen data: Ontario, Regional Municipality of Niagara,** Welland, 42.96198, −79.272590, 01 June 2018, coll: M. Pearce, black Lindgren funnel trap at ground level baited with general longhorn lure (CFIA 2).


**Distribution in Canada and Alaska: ON.**



***Lymantor alaskanus* Wood 1978, new to British Columbia.**


**Note:** Native to North America. Range: previously known only from Alberta and Alaska [74]. Host unknown. Note: subsequent collections from the same location in the same year yielded a further 505 individuals.

**Specimen data: British Columbia, Regional District of Fraser-Fort George, Prince George,** 53.91852, −122.73413, 15 May 2015, coll: A. Beuzer, black Lindgren funnel trap at ground level baited with *Monochamus* lure (CFIA-BC 5).

**Distribution in Canada and Alaska:** AK, **BC,** AB.


***Pityogenes bidentatus* (Herbst 1784) †, new to Canada and Ontario.**


**Note:** Native to Europe. Range: widespread throughout Europe, and reported from northern Asia, Japan, and Korea; adventive in eastern USA. Hosts: primarily *Pinus*. Usually breeds in fallen and cut branches but can also attack young trees in plantations and nurseries [44].

**Specimen data: Ontario, Regional Municipality of Niagara,** Port Colborne, 42.87705, −79.238390, 14 June 2018, coll: K. Cantera, black Lindgren funnel trap at ground level baited with *Monochamus* lure (CNC 1).


**Distribution in Canada and Alaska: ON.**



***Scolytus mali* (Bechstein 1805) †, new to British Columbia.**


**Note:** Native to the Palearctic. Range: widespread across Eurasia; northern Africa; adventive in eastern North America. First detected in NY in 1868 but then not again until 1933 [49]. In Canada, previously known from ON and QC [54]. Not previously known from western North America. This disjunct detection suggests either human-assisted movement or a new introduction. Hosts: *Malus, Prunus, Ulmus, Pyrus,* and *Sorbus* [49]. Phloeophagous, colonizes dying and weakened limbs.

**Specimen data: British Columbia, Metro Vancouver Regional District,** Richmond, 49.19323, −123.07320 and 49.19317 −123.07534, 23 June 2016, coll: T. Kimoto, green Lindgren funnel canopy trap baited with C6 + C8 ketol lure (CFIA-BC 1, CNC 1).

**Distribution in Canada and Alaska: BC,** ON, QC.


***Scolytus schevyrewi* Semenov 1902 †, new to Quebec.**


**Note:** First detected in North America in 1994 [75]. In Canada, previously known from ON to BC [56]. Host: *Ulmus* spp., possible vector of *Ophiostoma novo-ulmi*, the causative agent of Dutch elm disease [75].

**Specimen data: Quebec, TE de Longueuil,** Boucherville, 45.59787, −73.470120, 13 June 2019, coll: S. Lamontagne, green Lindgren funnel canopy trap baited with general longhorn lure (CFIA 1).

**Distribution in Canada and Alaska:** BC, AB, SK, MB, ON, **QC.**


***Trypodendron scabricollis* (LeConte 1868), new to Canada and Ontario.**


**Note:** Native to North America. Range: eastern North America. Host: *Pinus;* attacks the bole of unhealthy or injured trees; phloeophagous [51].

**Specimen data: Ontario, Elgin County,** St. Thomas, 42.83099, −81.177740, 24 April 2018, coll: P. Dolan, black Lindgren funnel trap at ground level baited with *Monochamus* lure (CFIA 3, CNC 4).


**Distribution in Canada and Alaska: ON.**



***Trypophloeus populi* Hopkins 1915, new to Quebec.**


**Note:** Native to North America. Range: Canada, USA, northern Mexico. Previously known from BC, AB, SK, MB, and NB [56]. Host: *Populus* spp.; attacks green bark of unhealthy trees [76].

**Specimen data: Quebec, TE de Sherbrooke,** Sherbrooke, 45.40162, −71.975520, 18 September 2017, coll: F. Paquette, black Lindgren funnel trap at ground level baited with general longhorn lure (CNC 1).

**Distribution in Canada and Alaska:** BC, AB, SK, MB, **QC,** NB.


***Xylechinus americanus* Blackman 1922, new to Newfoundland.**


**Note:** Native to North America. Range: eastern North America. In Canada, previously known from ON, QC, NB and NS [54]. Hosts: *Picea* spp. Generally attacks small shaded-out trees or lower, shaded, branches of dominant trees [51].

**Specimen data: Newfoundland, Placentia Bay,** Argentia, 47.28942, −53.980140, 17 July 2017, coll: S. Thordarson, black Lindgren funnel trap at ground level baited with *Monochamus* lure (CNC 1).

**Distribution in Canada and Alaska:** ON, QC, NB, NS, **NL.**


***Xylosandrus crassiusculus* (Motschulsky 1866) †, new to British Columbia and Quebec.**


**Note:** Native to East Asia. Range: Asia, Africa, South and Central America, recently several locations in Europe, eastern North America. Previously known from ON [54]. Hosts: Highly polyphagous; prefers hardwoods; has been reported on many fruit and nut crop trees [51,75]. Can colonize and kill small trees in nurseries and urban settings [77,78].

**Specimen data: British Columbia, Metro Vancouver Regional District,** North Vancouver, 49.30626, −123.03881, 07 Aug. 2012, coll: T. Graham, black Lindgren funnel trap at ground level baited with UHR ethanol & alpha-pinene (CFIA 1). **Quebec, MRC La Rivière-du-Nord,** Saint-Jérôme, 45.77905, −74.026470, 13 June 2017, coll: S. Lamontagne, black Lindgren funnel trap at ground level baited with *Monochamus* lure (CFIA 1).

**Distribution in Canada and Alaska: BC,** ON, **QC.**

## 4. Discussion

The inadvertent introduction of adventive and invasive insect species is a threat to the health, diversity, and productivity of forests in Canada and the world at large. Early detection of new invaders is essential for successful eradication or management of potential threats. Furthermore, documentation of new introductions and range expansions of insects is important for both a management approach and a biodiversity perspective. The forest pest surveillance activities conducted by the CFIA provide critical information to North American forest managers and to those interested in biodiversity. These activities are not static. In response to new information on the efficacy of various semiochemicals on detection of bark- and wood-boring beetles in traps, e.g., [23,24,25,26,27,55], CFIA changes its lure types every few years to improve trapping of target groups or to survey other beetle taxa. Additionally, results from research on the effects of trap location (e.g., canopy vs. understory, forest edge vs. interior) [79,80,81,82,83] on the efficacy of detecting bark- and wood-boring beetles are gradually being incorporated into CFIA’s surveillance programs. For example, the Eurasian *Scolytus mali*, was detected in an industrial zone in Richmond, BC in 2016 as the last author began to pilot the placement of green multiple-funnel traps in tree canopies. In 2018, CFIA started setting some green multiple-funnel traps in angiosperm canopies and now they currently are a minor proportion of all the traps set across Canada. Currently, the three CFIA offices in QC are piloting green canopy traps in various gymnosperms to determine the feasibility of setting these traps in coniferous canopies. In addition to *S. mali*, the new Canadian record of *A. maiche* and the new Quebec provincial record of *S. schevyrewi* were detected in green canopy traps.

The new records within this report can be categorized as either indigenous Canadian species that have been detected in new provinces (10 species), insects native to the USA that are new Canadian records (9 species), or species adventive to North America (9 species) (Table 2). The recent detection in Canada of several species native to the USA could be due to human-assisted movement of infested material, natural migration exacerbated by climate change, or both. It is also possible that some of the native American species were already present in Canada but were simply undetected prior to our surveys. Many of the records involve phloeophagous and xylomycetophagous beetles that could be transported in firewood, logs, or live plants. There are federal regulations on the movement of such materials into Canada but these regulations may not have been in place when these introductions occurred (i.e., delay between species establishment and detection) or people may not have complied with these regulations. Climate change may well have played a role in the northward expansion of these species; warming climate has already been linked to the northern expansion of the mountain pine beetle, *Dendroctonus ponderosae* Hopkins, in Canada [84,85].

Of the 9 new records of beetles not native to North America, 4 are new Canadian records while the other 5 are adventive beetles previously detected in other provinces. However, it is uncertain if the new Canadian records of *A. maiche*, *C. pelliculosum*, *P. bidentatus*, and *A. lepidii* arrived with goods imported from their native range or through introduction from the USA as all four species were previously known to occur there.

Rearing beetles from bolts of felled trees accounted for only two new records (*H. pruinosus*, *R. spretus*) but they also provide confirmation of their host plants. *Rhyncolus spretus* was reared from an *Acer* sp. bolt in New Westminster, BC. This weevil is native to western USA where it’s known to breed under the bark of *Corylus*, *Sambucus* and other angiosperms, and has been previously reared from *A. negundo* [65]. The CFIA rearing survey also previously detected the first Canadian records of *Trichoferus campestris* [14] and *Ernobius mollis* [86].

Twenty-four of the 31 new species records were detected in traps baited with one of the two lure combinations used from 2015 to 2021. The general longhorn lure (fuscumol, fuscumol acetate, UHR ethanol) accounted for 16 new records and the *Monochamus* lure (ipsenol, monochamol, UHR alpha-pinene and UHR ethanol) accounted for eight new records (Appendix A). Traps baited with the *Monochamus* lure were designed to primarily detect species that feed in conifers [79] so their detection of *H. fasciatus* and *X. crassiusculus,* which feed in *Fraxinus* spp. and a variety of angiosperms, respectively, was likely due to chance or the presence of ethanol [87]. Traps baited with the lure combination of C6 ketol + C8 ketol + UHR ethanol, deployed from 2012 to 2014 as well as in select sites in 2016, detected the new records of *N. m. mucronatus, S. mali*, and additional records of *H. opacus.* Detection of *N. m. mucronatus* in the ketol-baited traps was likely because 3-hydroxyhexan-2-one (C6 ketol) is a component of its aggregation pheromone [88]. Detection of the bark beetles in these traps may have been due to chance or the presence of UHR ethanol but it is also possible the presence of ketols affected attraction, e.g., adding racemic 3-hydroxyoctan-2-one to ethanol baited traps significantly increased catches of the bark beetle *Hypothenemus rotundicollis* (Eichhoff) [89].

One of the lessons learned from studies on the effect of semiochemicals [23,24,25,26,27,28,29,30,31,32,33,34], trap placement [79,80], and trap color [81,82,83] on detection of bark and woodboring beetles, and reinforced by the results of our study, is that no single trap-lure combination, trap color or trap height is effective at detecting the broad range of adventive species of Cerambycidae, Buprestidae and Curculionidae at risk of introduction via global trade. Using many different pheromones and host volatiles in multi-component lures has proved successful at detecting many species in port trapping surveys [90] but cost of baiting traps generally increases with the number of components, and with fixed budgets, there would be tradeoffs, e.g., fewer traps per site or fewer sites surveyed. The CFIA plans to maintain the number of sites and traps deployed per site, and to vary the multi-lure combinations used to bait traps every 2–3 years to increase the diversity of species potentially detected in traps. Black funnel traps placed in the understory have proven effective at detecting many species of Scolytinae and Cerambycidae but detect very few jewel beetles [79]. The CFIA has begun to introduce green canopy traps to more survey sites each year as these have proven far more effective at detecting jewel beetles, especially those in the genus *Agrilus,* e.g., [81,91,92].

Since 2011, true weevils (Curculionidae) accounted for the largest proportion of new records (28 of 31 records or 90%) and 18 of these were in the subfamily Scolytinae, bark and ambrosia beetles. Jewel beetles (Buprestidae), longhorned beetles (Cerambycidae) and bostrichid powder-post beetles (Bostrichidae) were represented by one new record each. Table 2 shows the updated distribution of all these species in Canada.

The bostrichid beetle, *Amphicerus cornutus* is native and widespread in the USA, Mexico, and South America, so it is perhaps not surprising that it has been detected in BC. If human-assisted movement was responsible for its occurrence in Canada, it is curious that it was first detected in Vernon which has a much smaller population size than greater Vancouver and Kelowna, especially as these regions are consistently surveyed by CFIA.

*Neoclytus m. mucronatus* is indigenous to eastern North America [56] and was the lone cerambycid record. This disjunct detection on the west coast combined with the rarity of its host trees (*Celtis, Carya*) in Surrey BC where it was detected and no additional detections in CFIA’s annual trapping surveys in BC since 2012 suggests this may be an interception of a beetle that emerged from infested material (possibly firewood) transported to the area. Additional trapping surveys in neighboring municipalities would be necessary to determine its prevalence and whether it has established in the greater Vancouver area.

Even though the trapping survey was designed to target wood-boring insects, other fauna were also caught such as *Aulacobaris lepidii* and *Mecinus janthinus*. The former is a Palearctic species that is adventive to northeastern USA and its hosts are Brassicaceae (mustards or crucifers). *Mecinus janthinus* is native to Europe and was intentionally introduced into North America as a biological control agent of toadflax, which is also of European origin. It had been previously discovered in QC and NS; its detection in ON is not surprising given that toadflax also occurs in this province.

Detection of adventive forest insects in annual survey trapping surveys, combined with pest risk assessments, is a critical part of CFIA’s strategy for protecting Canada’s forests. Fortunately, none of the adventive species detected in surveys between 2011 and 2021 was considered a significant risk to Canada’s forest or forest products, and none were designated as quarantine pests. However, some of the adventive Scolytinae species that primarily attack stressed trees, such as *X. crassisusculus,* may become more damaging in the future, if larger areas of forests become stressed due to more severe or frequent extreme weather events associated with a warming climate [93]. The new species records reported herein enrich our knowledge of Canada’s biodiversity and are an indication of the value of the surveillance activities conducted by the CFIA. They also indicate a level of change occurring in Canadian forests, and the need for continued monitoring of changes in our global forest insect fauna.

## Figures and Tables

**Table 1 insects-13-00708-t001:** Abbreviations of provinces used in the text.

**AK** Alaska	**QC** Quebec
**BC** British Columbia	**NB** New Brunswick
**AB** Alberta	**PE** Prince Edward Island
**SK** Saskatchewan	**NS** Nova Scotia
**MB** Manitoba	**NL & LB** Newfoundland and Labrador *
**ON** Ontario	

* The island of Newfoundland and Labrador are each treated separately under the current **Distribution in Canada and Alaska**.

**Table 2 insects-13-00708-t002:** Distribution in Canada, with new records highlighted (**X**). Species name in bold denotes a new Canadian record, and species with † are adventive to North America. None of the species have been recorded in Nunuvut, Yukon or Northwest Territory so these are not listed in the table.

Family	Species	BC	AB	SK	MB	ON	QC	NB	NS	PE	NL
Buprestidae	** *Xenomelanophila miranda* **	**X**									
Cerambycidae	*Neoclytus m. mucronatus*	**X**				x		x			
Bostrichidae	** *Amphicerus cornutus* **	**X**									
Curculionidae	*Mecinus janthinus* †	x	x			**X**	x		x		
Curculionidae	***Aulacobaris lepidii*** †					**X**					
Curculionidae	*Buchananius striatus*					**X**	x				
Curculionidae	** *Cylindrocopturus binotatus* **					**X**					
Curculionidae	*Himatium errans*					**X**	x	x	x		
Curculionidae	*Phloeophagus canadensis*	x	x	x	x	**X**	x	x			
Curculionidae	** *Rhyncolus spretus* **	**X**									
Curculionidae	** *Stenomimus pallidus* **					**X**					
Curculionidae	** *Tomolips quercicola* **					**X**					
Curculionidae	*Strophosoma melanogrammum* †	x				X	x	**X**	x	x	x
Curculionidae	*Conotrachelus aratus*					**X**	x				
Curculionidae	***Anisandrus maiche*** †					**X**	**X**				
Curculionidae	** *Cnesinus strigicollis* **					**X**					
Curculionidae	***Cyclorhipidion pelliculosum*** †					**X**	**X**				
Curculionidae	*Hylastes opacus* †	x				x	x	x	x		
Curculionidae	*Hylesinus fasciatus*					X	**X**				
Curculionidae	*Hylesinus pruinosus*					X	**X**				
Curculionidae	** *Hypothenemus interstitialis* **					**X**					
Curculionidae	*Lymantor alaskanus*	**X**	x								
Curculionidae	***Pityogenes bidentatus*** †					**X**					
Curculionidae	*Scolytus mali* †	**X**				X	x				
Curculionidae	*Scolytus schevyrewi* †	x	x	x	x	X	**X**				
Curculionidae	** *Trypodendron scabricollis* **					**X**					
Curculionidae	*Trypophloeus populi*	x	x	x	x		**X**	x			
Curculionidae	*Xylechinus americanus*					X	x	x	x		**X**
Curculionidae	*Xylosandrus crassiusculus* †	**X**				X	**X**				

## Data Availability

Voucher specimens supporting the data on species records in this paper have been deposited in either the Canadian National Collection of Insects in Ottawa, Canada (CNC) or the Collection of the Canadian Food Inspection Agency, Ottawa, Canada, or both, as noted in the body of the paper.

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
