# Peer review of "New Canadian and Provincial Records of Coleoptera Resulting from Annual Canadian Food Inspection Agency Surveillance for Detection of Non-Native, Potentially Invasive Forest Insects"

_insects, 2022, doi:10.3390/insects13080708_

Round 1
Reviewer 1 Report
This is a very good, well written paper, with interesting and valuable information on foreign insects caught in traps and reared from infested logs that are new for Canada, representing real or potential threats for Canadian forests. These surveys are important in early detection of these insects so that they can be eradicated before establishing permanently. I have the following comments:
Line 9: Why Canadian underlined and in bold?
Lines 22 and 25. The question arises why only focus on forests and wood, while such surveys are also important for agriculture generally. Therefore the focus narrower focus should be reflected in the title of the manuscript so that you do not disappoint a reader (like me) expecting a more general survey from the title.
63. Full stop.
159. Should names not be written out or an indication given where they can be found?
175. Align above.
184. Not bold. Table 1. Check alignment.
185-436. Is this section part of the Table? If so, I consider the wasteful double spacing and general presentation unacceptable and unsuitable. Consider making it a subsection of the Results. The Note, Specimen and Distribution should not be broken up in my view. And what do the figures such as 50.22030,-119.30720 indicate?
568. Full stop after J.
617.Entomol in italics.
727.Journal of Pest Science as in 720
Author Response
Please see attachment in which we respond to comments by Reviewers 1, 2 and 3

Reviewer 2 Report
A totally straightforward and useful paper. It is very helpful when the results of multi-year surveys make it into the literature, for exactly the reasons the authors state in the intro/discussion. I appreciate especially calling out the fact that the green canopy funnels may have a role in expanding the taxonomic breadth of these types of surveys. I have made a few comments in the attached pdf about minor suggestions for wording and presenting some of the results... but they are seriously minor. If this paper was published as is it would be totally fine. Please let me know if the comments in the PDF are not visible.

Author Response
Please see attachment in which we respond to comments by Reviewers 1, 2 and

Reviewer 3 Report
This paper reports on a 11-yr survey for detecting potentially damaging non-native bark and wood boring insects throughout Canada. The findings are important and valuable to document both the survey efforts fully described and the results. Lastly, what the implications of these findings mean and what follow-up efforts are suggested…..This is what the paper should have been.
But some of these items are not included.
Regarding the trapping efforts, the narrative is confusing. With a little effort , a table could be created that lists which traps were deployed, where, when, and what general groups of insects they were targeting. Fully documenting the survey effort over this 11-yr period is as critical as the results. Lets get a fuller picture of what CFIA did. A table would go a long way to helping the reader examine the overall effort. For example there is a description of different lures targeting different insects (L. 120-135). Were all the different lures place in one trap, or were traps separated and differentiated to attract different subfamilies of insects? It is not clear to me how the traps were deployed in regards to this. Additionally, I’m curious, how many total traps were deployed annually? Maybe a map showing all the trapping locations in Canada (by targeted subfamilies) would help with this in addition to the table or instead.
Following the results section, the Discussion falls shot on addressing what it means to have detected these new species. Are any of these considered problematic? If so what is to be done as a follow-up for finding them? Lastly, where does the program go from here? What was learned and where does it take CFIA survey efforts into the future?
One other item.
L 113-114. I know the Lindgren funnel traps are fairly ubiquitous for bark and wood boring insects, but for those not familiar with them, it would be nice to include a photo or at least a citation.
Author Response
Please see previous uploaded document "Response to reviewers" for response to Reviewer 3. Here we add an annotated pdf with our responses to comments made by one of the reviewers in an annotated pdf.

Round 2
Reviewer 3 Report
The authors addressed all of this reviewer's comments to my satisfaction.
Author Response
thanks for your assistance again.